# Decoding Mental Effort in a Quasi-Realistic Scenario: A Feasibility Study on Multimodal Data Fusion and Classification

**DOI:** 10.3390/s23146546

**Published:** 2023-07-20

**Authors:** Sabrina Gado, Katharina Lingelbach, Maria Wirzberger, Mathias Vukelić

**Affiliations:** 1Experimental Clinical Psychology, Department of Psychology, Julius-Maximilians-University of Würzburg, 97070 Würzburg, Germany; sabrina.gado@uni-wuerzburg.de; 2Applied Neurocognitive Systems, Fraunhofer Institute for Industrial Engineering IAO, 70569 Stuttgart, Germany; mathias.vukelic@iao.fraunhofer.de; 3Applied Neurocognitive Psychology Lab, Department of Psychology, Carl von Ossietzky University, 26129 Oldenburg, Germany; 4Department of Teaching and Learning with Intelligent Systems, University of Stuttgart, 70174 Stuttgart, Germany; maria.wirzberger@iris.uni-stuttgart.de; 5LEAD Graduate School & Research Network, University of Tübingen, 72072 Tübingen, Germany

**Keywords:** mental effort, machine learning, multimodal physiological signals, sensor fusion, neuroergonomics, human–machine interaction

## Abstract

Humans’ performance varies due to the mental resources that are available to successfully pursue a task. To monitor users’ current cognitive resources in naturalistic scenarios, it is essential to not only measure demands induced by the task itself but also consider situational and environmental influences. We conducted a multimodal study with 18 participants (nine female, M = 25.9 with SD = 3.8 years). In this study, we recorded respiratory, ocular, cardiac, and brain activity using functional near-infrared spectroscopy (fNIRS) while participants performed an adapted version of the warship commander task with concurrent emotional speech distraction. We tested the feasibility of decoding the experienced mental effort with a multimodal machine learning architecture. The architecture comprised feature engineering, model optimisation, and model selection to combine multimodal measurements in a cross-subject classification. Our approach reduces possible overfitting and reliably distinguishes two different levels of mental effort. These findings contribute to the prediction of different states of mental effort and pave the way toward generalised state monitoring across individuals in realistic applications.

## 1. Introduction

In everyday life, we constantly face situations demanding high stakes for maximum gains; for instance, to succeed in rapidly acquiring complex cognitive skills or making decisions under high pressure. Thereby, a fit between personal skills and the task’s requirements determines the quality of outcomes. This fit is vital, especially in performance-oriented contexts such as learning and training, safety-critical monitoring, or high-risk decision-making. A person’s performance can be affected by several factors: (1) level of experience and skills, (2) current physical conditions (e.g., illness or fatigue), (3) current psychological conditions (e.g., stress, motivation, or emotions), or (4) external circumstances (e.g., noise, temperature, or distractions; Hart and Staveland [1], Young et al. [2]).

To reliably quantify the mental effort during a particular task, different measures can be used: (1) behavioural (i.e., performance-based), (2) subjective, and (3) neurophysiological measures [3,4,5]. While performance can be inspected by tracking the user’s task-related progress, the actual pattern of invested cognitive resources can only be derived by measuring brain activity with neuroimaging techniques. Coupled with sophisticated signal processing and machine learning (ML), advances in portable neuroimaging techniques have paved the way for studying mental effort and its possible influences from a neuroergonomic perspective [6,7]. Recently, functional near-infrared spectroscopy (fNIRS) has been used to study cognitive and emotional processes with high ecological validity [6,8,9,10]. fNIRS is an optical imaging technology allowing researchers to measure local oxy-haemoglobin (HbO) and deoxy-haemoglobin (HbR) changes in cortical regions. Higher mental effort is associated with an increase in HbO and a decrease in HbR in the prefrontal cortex (PFC) [11,12,13]. The PFC is crucial for executive functions like maintaining goal-directed behaviour and suppressing goal-irrelevant distractions [14,15]. In addition to changes in the central nervous system, an increased mental effort also leads to changes in the autonomic nervous system. The autonomic nervous system, as part of the peripheral nervous system, regulates automatic physiological processes to maintain homeostasis in bodily functioning [16,17]. Increased mental effort is associated with decreased parasympathetic nervous system activity and increased sympathetic nervous system activity [18,19,20]. Typical correlates of the autonomic nervous system for cognitive demands, engagement or mental effort are cardiac activity (e.g., heart rate and heart rate variability), respiration (rate, airflow, and volume), electrodermal activity (skin conductance level and response), blood pressure, body temperature, and ocular measures like pupil dilation, blinks, and eye movements [7,19,21,22,23,24].

Not surprisingly, all these measures are, thus, often used as a stand-alone indicator for mental effort (i.e., in a *unimodal approach*). However, a *multimodal approach* has several advantages over using only one measure. It can compensate for specific weaknesses and profit from the strengths of the different complementary measurement methods (performance, subjective experience as well as neuro- and peripheral physiological measures) [25,26,27]. For instance, (neuro-)physiological measures can be obtained without imposing an additional task [16] and allow for capturing cognitive subprocesses involved in executing the primary task [28]. A multimodal approach, hence, provides a more comprehensive view of (neuro-)physiological processes related to mental effort [4,5,25,29], as it can capture both central and peripheral nervous system processes [21,27]. However, fusing data from different sources remains a major challenge for multimodal approaches. ML methods provide solutions to compare and combine data streams from different measurements. ML algorithms are becoming increasingly popular in computational neuroscience [30,31]. The rationale behind these algorithms is that the relationship between several input data streams and a particular outcome variable, e.g., mental effort, can be estimated from the data by iteratively fitting and adapting the respective models. This allows for data-driven analyses and provides ways to exploratorily identify patterns in the data that are informative [32].

Data-driven approaches can also be advantageous in bridging the disparity between laboratory research and real-world applications. For instance, when specific temporal events (such as a stimulus onset) or the brain correlates of interest, are not precisely known. In contrast to traditional laboratory studies that typically rely on simplified and artificial stimuli and tasks, a naturalistic approach seeks to emulate, to some extent, the intricacy of real-world situations. Hence, these studies can provide insights into how the brain processes information and responds to complex stimuli in the real world [33].

Real-world settings are usually characterised by multiple situational characteristics, including concurrent distractions that affect the allocation of attentional and cognitive resources [34]. According to the working memory model by Baddeley and Hitch [35], performance is notably diminished when distractions deplete resources from the same modality as the primary task. However, Soerqvist et al. [36] propose the involvement of cognitive control mechanisms that result in reduced processing of task-irrelevant information under higher mental effort. To uphold task-relevant cognitive processes, high-level cortical areas, particularly the PFC, which govern top-down regulation and executive functioning, suppress task- or stimulus-irrelevant neural activities by inhibiting the processing of distractions [28]. Consequently, the effects of distractors are mitigated. In light of these considerations, understanding the capacity of a stimulus to capture attention in a bottom-up manner, known as salience, emerges as a crucial aspect. A salient stimulus has the potential to disrupt top-down goal-oriented and intentional attention processes [37] and to impair performance in a primary task [38,39,40]. Previous studies found that irrelevant, yet intelligible speech exerts such disruptive effects on participants’ performance in complex cognitive tasks [41,42]. Consequently, intelligible speech might heighten the salience of a distracting stimulus. Moreover, further studies revealed that the emotional intensity and valence of a stimulus also play a role in influencing its salience [37,43]. Despite their detrimental impact on performance, people frequently experience such salient distractions (such as verbal utterances from colleagues) at work, even in highly demanding safety-relevant tasks. Therefore, gaining an understanding of the underlying cognitive processes in naturalistic scenarios and identifying critical moments that lead to performance decreases in real-world settings are crucial research topics in the field of neuroergonomics.

To decode and predict cognitive states, most research so far focused on subject-dependent classification. These approaches face the challenge of high inter-individual variability in physiological signals when generalising the model to others [44]. Recently, pioneering efforts have been made to develop cross-subject models that overcome the need for subject-specific information during training [45,46]. Solutions to address the challenge of inter-individual variability [47] are crucial for the development of “plug and play” real-time state recognition systems [48] as well as the resource-conserving exploitation of already available large datasets without time-consuming individual calibration sessions. Taking into account the aforementioned considerations and research agenda concerning the decoding of mental effort in naturalistic scenarios, we conducted a feasibility study and developed an ML architecture to decode mental effort across subjects from multimodal physiological and behavioural signals. We used a monitoring task simulating typical work tasks of air traffic controllers. This adapted version of the warship commander task induces mental effort based on a combination of attentional and cognitive processes, such as object perception, object discrimination, rule application, and decision-making [49]. To create a complex close-to-naturalistic scenario, three emotional types of auditory speech-based stimuli with neutrally, positively, and negatively connotated prosody were presented during the task as concurrent distractions [50]. Concurrently, performance-based, brain-related as well as peripheral physiological signals associated with mental effort were recorded.

We hypothesised that a well-designed multimodal voting ML architecture is preferable compared to a classifier based on (a) only one modality (unimodal approach) and (b) a combined, unbalanced feature set of all modalities. We expected that a multimodal voting ML model is capable of predicting subjectively experienced mental effort induced by the task itself but also by the suppression of situational auditory distractions in a complex close-to-realistic environment. Thus, we first investigated whether a combined prediction of various ML models is superior to the prediction of a single model (RQ1) and, second, we explored whether a multimodal classification that combines and prioritises the predictions of different modalities is superior to a unimodal prediction (RQ2). Furthermore, our approach enables a systematic evaluation of the unimodal and multimodal models, assessing their suitability and informativeness of each modality in decoding mental effort. This knowledge provides researchers in the fields of Human Activity Recognition and Behaviour Recognition with references for selecting suitable sensors, as well as a validated multimodal experimental design and ML processing pipeline [51].

## 2. Materials and Methods

### 2.1. Participants

Interested volunteers filled in a screening questionnaire that checked eligibility for study participation and collected demographic characteristics. Participants between the ages of 18 and 35 and with normal or corrected-to-normal vision were included in the study. Interested volunteers were excluded if they had insufficient knowledge of the German language or limited colour vision, as these factors could impede their ability to perform the tasks. Additionally, pregnant women, individuals indicating precarious alcohol or drug consumption, and those reporting mental, neurological, or cardiovascular diseases were not included. Due to the data collection period in June 2021 coinciding with the COVID-19 pandemic, individuals belonging to the risk group for severe COVID-19 disease, as defined by the Robert Koch Institute, were also not invited to the laboratory. The final sample consisted of 18 participants (nine female, three left-handed, mean age of 25.9 years, SD=3.8, range = 21–35 years) who were all tested individually. Before their participation, they signed an informed consent according to the recommendations of the Declaration of Helsinki and received monetary compensation for their voluntary participation. The study was approved by the ethics committee of the Medical Faculty of the University of Tübingen, Germany (ID: 827/2020BO1).

### 2.2. Experimental Task

Participants performed an adapted version of a warship commander task (WCT [52]; adapted by Becker et al. [49]). The WCT is a quasi-realistic navy command and control task designed as a basic analogue to a Navy air warfare task [53]. It is suitable to investigate various cognitive processes of human decision-making and action execution [53]. Here, we used a non-military safety-critical task, where participants had to identify two different flying objects on a simulated radar screen around an airport. Objects included either registered drones (neutral, non-critical objects), or non-registered (critical) drones. They had to prevent the non-registered drones, potentially being a safety issue, from entering the airport’s airspace. Non-registered drones entering pre-defined ranges close to the airport had to be first warned and then repelled in the next step. A performance score was computed based on participants’ accuracy and reaction time. See Becker et al. [49], for a more detailed description of the scoring system, and see Figure 1 for an overview of the interface.

During the task, we presented vocal utterances, either spoken in a happy, angry, or neutral way from the Berlin Database of Emotional Speech (Emo-DB [50]). These utterances were combined into different audio files, each one minute long, with speakers and phrases randomly selected and as little repetition as possible within each file. We also included a control condition where no auditory distraction was presented. The task load was manipulated by implementing two difficulty levels in the WCT (low and high). This resulted in a 2 × 4 design with eight experimental conditions. Participants completed two rounds of all conditions in the experiment. Before the respective round, a resting state measurement was conducted (30 s). Each round then consisted of eight blocks, each comprising three 60-s trials of the same experimental condition. The task load condition (operationalised with the difficulty level) was alternated across blocks. Half of the subjects started with a high task load and the other half with a low task load block. Similarly, the concurrent emotional condition (operationalised with different auditory distractions) was randomised and sampled without replacement. Before each block, except for the first, participants completed a baseline condition trial with a very low difficulty level where they had to track six objects, of which three were non-registered drones. In the low task load condition, participants had to track 12 objects, of which six were non-registered drones. In the high task load condition, they had to track 36 objects, of which 17 were non-registered drones. We used different emotional audio files for the trials in one block. Before and after the whole experiment, as well as after each experimental block, participants filled in questionnaires. See Figure 2 for a schematic representation of the whole experimental procedure. Overall, the experiment lasted approximately 120 min, including 30 min of preparation time of the used measurement devices and calibration procedures.

### 2.3. Data Collection

#### 2.3.1. Questionnaires

Subjectively perceived mental effort and affective states were assessed after each experimental block. We used the NASA TLX effort and frustration subscales [1], EmojiGrid [54], and categorical Circumplex Affect Assessment Tool (CAAT [55]). After the experiment, participants answered questionnaires regarding personal traits that might have influenced their performance and behaviour during the study. These questionnaires comprised the short version of the German Big Five Inventory (BFI-K [56]), the German State-Trait-Anxiety Inventory (STAI [57]), the Attention and Performance Self-Assessment (APSA [58]) and the German language version of the Barratt Impulsiveness Scale-11 (BIS [59]). Here, we only used the NASA TLX ratings of mental effort for labelling the (neuro-)physiological data in the ML classification. The other subjective measures were not of interest in this analysis.

#### 2.3.2. Eye-Tracking, Physiology, and Brain Activity

The ocular activity was recorded with the screen-based Tobii Pro Spectrum eye-tracking system, which provides gaze position and pupil dilation data at a sampling rate of 60 Hz. To capture changes in physiological responses, participants were wearing a Zephyr BioHarness™ 3 belt recording electrocardiographic (ECG), respiration, and temperature signals at a sampling rate of 1 Hz. Here, we used automatically computed, aggregated scores for the heart rate, heart rate variability, and respiration rate and amplitude from the device. Physiological as well as behavioural measures were recorded using the iMotions Biometric Research Platform software. Participants’ brain activity was recorded with a NIRx NIRSport2 system, which emits light at two wavelengths, 760 and 850 nm. Data were collected with the Aurora fNIRS recording software at a sampling rate of 5.8 Hz. To capture regions associated with mental effort, 14 source optodes and 14 detector optodes were placed over the prefrontal cortex [12,60] using the fNIRS Optodes’ Location Decider (fOLD) toolbox [61] (Figure 3, for the montage). Event triggers from the experimental task were sent to iMotions and Aurora using TCP protocols and Lab Streaming Layer (LSL). Signals from the different recording and presentation systems were temporally aligned offline after the data collection.

### 2.4. Data Preprocessing and Machine Learning

Data preprocessing and ML analyses were performed with custom-written scripts in R (version 4.1.1) and Python™ (version 3.8). Continuous raw data streams were cut into non-overlapping 60-s intervals starting at the onset of each experimental trial (Figure 2). Before feeding the data into the classification pipeline, we applied the following data cleaning and preprocessing steps per modality.

#### 2.4.1. Preprocessing of Eye-Tracking Data

Continuous eye tracker data were preprocessed using the eyetrackingR package in R [62]. Missing values were linearly interpolated and 855 trials with a length of 60 s (on average 47.5 trials per subject, SD=0.9) were extracted. Next, we used the validity index to remove non-consistent data segments from further analysis. The index is provided by the eye tracker and indicates samples in which the eye tracker did not recognise both pupils correctly (“track loss”). A total of 17 trials (1.99%) with a track loss proportion greater than 25% were removed, and 838 trials were left to extract fixations and pupil dilation (on average 46.6 trials per subject, SD=2.4). For the preprocessing of the pupil dilation data, we used the PupillometryR R-package [63]. First, we calculated a simple linear regression of one pupil against the other and vice versa, per subject and trial to smooth out small artefacts [64]. Afterwards, we computed the mean of both pupils and filtered the data using the median of a rolling window with a size of 11 samples. To control for the variance of pupil sizes between participants, we applied a subject-wise z-score normalisation of pupil dilation. For the computation of fixations, we used the saccades R-package [65]. We obtained fixations for 565 trials (on average 31.4 trials per subject, SD=1.2). To control for the variance between participants, we also computed z-scores of the number and the duration of fixations separately for each subject.

#### 2.4.2. Preprocessing of Physiological Data

Epoching in non-overlapping 60-s time windows from the electrocardiographic raw data resulted in 832 trials (on average 46.2 trials per subject, SD=1.3). We applied a correction for the between-participant variance identical to the one described for the eye-tracking data using z-score normalisation.

#### 2.4.3. Preprocessing of fNIRS Data

We used the libraries MNE-Python [66] and its extension MNE-NIRS [67] and guidelines from Yücel et al. [68] to preprocess the fNIRS data. First, we converted the raw data into an optical density measure. A channel pruning was applied using the scalp-coupling index for each channel which is an indicator of the quality of the connection between the optodes and the scalp and looks for the presence of a prominent synchronous signal in the frequency range of cardiac signals across the photo-detected signals [69]. Channels with a scalp-coupling index below 0.5 were marked as bad channels. We further applied a temporal derivative distribution repair accounting for a baseline shift and spike artefacts [70]. Channels marked as bad were interpolated, with the nearest channel providing good data quality. Afterwards, a short-separation regression was used, subtracting short-channel data from the standard long-channel signal to correct for systemic signals contaminating the brain activity measured in the long-channel [68,71]. Next, the modified Beer–Lambert Law was applied to transform optical density into HbO and HbR concentration changes [72] with a partial pathlength factor of 6 [66]. Data were filtered using a fourth-order zero-phase Butterworth bandpass filter to remove instrumental and physiological noise (such as heartbeat and respiration; cut-off frequencies: 0.05 and 0.7 Hz; transition bandwidth: 0.02 and 0.2 Hz). HbO and HbR data was cut into epochs with a length of 60 s and channel-wise z-scored normalised. In total, 730 trials were obtained for the analysis (on average 40.6 trials per subject, SD=9.6).

#### 2.4.4. Feature Extraction

Our feature space comprised brain activity, physiological, ocular and performance-related measures. Table 1 gives an overview of the included features per subject and trial for each modality. We extracted the features of the fNIRS data using the mne-features package [73].

Appendix A provide exploratory analyses of the distribution and relationship between behavioural, heart activity, respiration, ocular measures, and the NASA TLX questionnaire scale effort during low and high subjective load. Appendix A compare the grand average of the behavioural and physiological measures as well as single fNIRS channels of the prefrontal cortex using bootstrapping with 5000 iterations and 95% confidence intervals (CI) during low and high subjective load.

#### 2.4.5. Ground Truth for Machine Learning

Our main goal was to predict the mental effort experienced by an individual using ML and training data from other subjects (e.g., [74,75]). Since the experimentally manipulated task load was further influenced by situational demands (e.g., inhibiting task-irrelevant auditory emotional distraction), the perceived mental effort might not be fully captured by the experimental condition. Therefore, we explored two approaches to operationalise mental effort as a two-class classification problem: First, based on self-reports using the NASA TLX effort subscale, and second, based on the experimental task load condition.

For the mental effort prediction based on subjective perception, we performed a subject-wise median split and categorised values above the threshold as “high mental effort” and below as “low mental effort”. Across all subjects, we had a mean median-based threshold of 3.8 (SD=3.2, scale range = 0–20) leading to an average of 23.8 trials per subject with low mental effort (SD=6.6, range = 12–39) and 14.5 trials per subject with high mental effort (SD=6.2, range = 3–21; see Appendix A for a subject-wise distribution of the classes).

In addition, we performed a subject-wise split at the upper quartile of the NASA TLX effort subscale. The upper (or third) quartile is the point below which 75% of the data lies. We introduced this data split to also investigate the prediction and informative features of extremely high perceived mental effort, which may indicate cognitive overload. By performing a quartile split, we had a mean threshold of 6.1 (SD=4.3, scale range = 0–20) across all relevant subjects (excluding subjects 5 and 9 which did not show enough variation to identify these two classes) with an average number of 30.9 low mental effort trials per subject (SD=8.0, range = 16–39) and 6.6 high mental effort trials per subject (SD=2.3, range = 3–9; see Appendix A for a subject-wise distribution of the classes).

At last, we compared the prediction of subjectively perceived mental effort with a prediction of the mental effort induced by the task, that is the experimental condition (“high task load” vs. “low task load”; see Appendix A for a subject-wise comparison of perceived mental effort dependent on the experimental load condition). The comparison allows to further control for confounding effects that are typical for self-reports, e.g., consistency effects or social desirability effects.

#### 2.4.6. Model Evaluation

We fitted six ML approaches: (1) Logistic Regression (LR), (2) Linear Discriminant Analysis (LDA), (3) Gaussian Naïve Bayes Classifier (GNB), (4) K-Nearest Neighbour Classifier (KNN), (5) Random Forest Classifier (RFC), and (6) Support Vector Machine (SVM). They were implemented using the scikit-learn package (version 1.0.1; [76]). Figure 4 shows a schematic representation of our multimodal classification scheme and cross-subject validation procedure using multiple randomised grid search operations.

For the cross-subject classification, we used a leave-one-out (LOO) approach where each subject served as a test subject once (leading to 18 “outer” folds). With this 18-fold cross-subject approach, we simulate a scenario where a possible future system can predict an operator’s current mental effort during a task without having seen any data (e.g., collected in a calibration phase) from this person before. This has the advantage that the model learns to generalise across individuals and allows the exploitation of already collected datasets from a similar context as training sets.

Our multidimensional feature space consisted of four modalities: (1) brain activity, (2) physiological activity, (3) ocular measures, and (4) performance measures. All features were z-standardised (Figure 4). This scaling ensured, that for each feature the mean is zero and the standard deviation is one, thereby, bringing all features to the same magnitude. We then trained the six classifiers (LR, LDA, GNB, KNN, RFC, and SVM) separately for each modality. Hyperparameters for each classifier were optimised by means of a cross-validated randomised grid search with a maximum number of 100 iterations and a validation set consisting of either one or two subjects. We tested both sizes of the validation set to find an optimal compromise between the robustness of the model and the required computing power. While cross-validation with two subjects counteracts the problem that the models highly adapt to an individual’s unique characteristics, cross-validation with only one subject leads to a lower number of necessary iterations and a computationally more efficient approach. Due to our cross-subject approach, the selected hyperparameters varied for each predicted test subject.

Afterwards, we combined these classifiers using a voting classifier implemented in the mlxtend package (version 0.19.0 [77]). The ensemble classifier makes predictions based on aggregating the predictions of the previously trained classifiers by assigning weights to each of them. Here, we are interested in whether an ensemble approach achieves higher prediction accuracy than the best individual classifier in the ensemble. An ensemble approach has the advantage that, even if each classifier is a weak learner (meaning it does only slightly better than random prediction), the ensemble could still be a strong learner (achieving high accuracy). The voting either follows a “soft” or a “hard” voting strategy. While hard voting is based on a majority vote combining the predicted classes, soft voting considers the predicted probabilities and selects the class with the highest probability across all classifiers. The weights, as well as the voting procedure (soft or hard voting), were optimised using a third cross-validated randomised grid search with a maximum number of 100 iterations. We restricted the weights to a maximum value of 2 (range = 0–2).

With this procedure, we were able to compare the predictions of the single unimodal classifiers to a weighted combination of all classifiers of one modality.

For the multimodal approach, the voting predictions of each modality were combined into a final multimodal prediction of mental effort using a second voting classifier. This second voting classifier also assigned weights to the different modality-specific classifiers and was optimised in the same manner as the unimodal approach. We report the average F_1_ score and a confusion matrix of the training set and the test subject to evaluate model performance. The F_1_ score can be interpreted as a weighted average or “harmonic mean” of precision and recall (1—good to 0—bad performance). Precision refers to the number of samples predicted as positive that are positive (true positives). Recall measures how many of the actual positive samples are captured by the positive predictions (also called sensitivity). The F_1_ score balances both aspects – identifying all positive, i.e., “high mental effort” cases, but also minimising false positives.

To compare the classification performance of different models, we calculated the bootstrapped mean and its CI over cross-validation folds with 5000 iterations per classification model. We corrected the 95% CI for multiple comparisons using the Bonferroni Method. Significant differences can be derived from non-overlapping notches of the respective boxes, which mark the upper and lower boundaries of bootstrapped 95% CI of the mean F_1_ score. The upper CI limit of a dummy classifier represents an empirical chance level estimate (dashed grey line in all subplots of Figure 5). A dummy classifier considers only the distribution of the outcome classes for its prediction. For a prediction to be better than chance (at a significance level below 0.05), the bootstrapped mean of a classifier must not overlap with this grey line [78]. For a significance level below 0.01, the lower CI boundary of a classifier’s mean must not overlap with this grey line [78]. This criterion can also be applied when statistically comparing different models, regardless of the chance level.

## 3. Results

We compare the results for a mental effort prediction based on a subject-wise (1a) median and (1b) upper quartile split of the Nasa TLX effort scale as well as based on the (2) experimentally induced task load. Further, we compare two sizes of the validation set (one subject and two subjects).

### 3.1. Unimodal Predictions

The performance of the different modalities and classifiers is visualised in Figure 5. We do not see substantially better performance when using a larger validation set of two subjects, neither for the median split (compare Figure 5 and Appendix A) nor for the upper quartile split (compare Appendix A) or the prediction of the experimentally induced task load (Appendix A). We will, therefore, focus on the models fitted with a validation set of one subject, as this is more time- and resource-efficient. In this case, we estimated the CI’s upper boundary of the mean empirical chance level for predicting subjectively perceived mental effort to be 0.444 (*M*
=0.368, 95% CI [0.284; 0.444]). This estimate now serves as a reference for determining significant performance above the chance level.

Figure 5 and Table 2 show the performance in a median-split-based unimodal approach (Figure 5A,B,D,E) as well as for the multimodal approach (Figure 5C; elaborated on in Section Multimodal Predictions). Regarding the unimodal classifications, we see the highest predictions of the subjectively perceived mental effort for performance data (Figure 5E compared with ocular, physiological, or brain activity measures; Figure 5A,B,D; see Table 2). Except for the performance-based model, we observe overfitting indicated by the large deviation between training and test performance (Figure 5A,B,D). None of the brain activity-based models performs significantly better than the estimated chance level in the test data set (Figure 5A,G, Table 2). When examining the single classification models within each modality, the KNN, RFC, and SVM were more likely to be overfitted, as seen by the good performance in the training set but a significantly worse performance for the test subject. We combined the different classifiers using a voting classifier, of which we ascertained the voting procedure (soft vs. hard voting) and the weights with a randomised grid search. See Figure 6 for an overview of the selected voting procedures and the allocated weights per modality.

**Figure 5 sensors-23-06546-f005:**
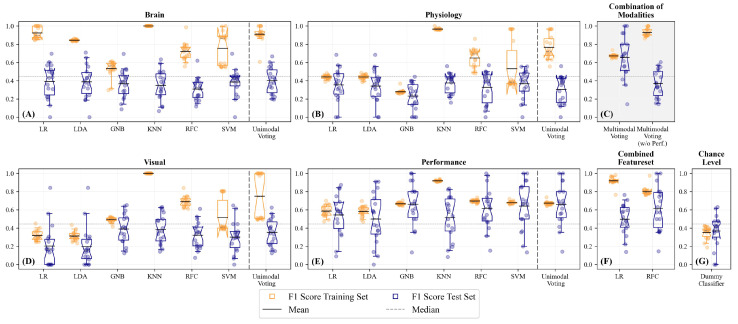
Prediction of the subjectively perceived mental effort based on a median split; validation set: N=1. Classifiers’ performance based on (**A**) fNIRS data, (**B**) physiological data (heart rate, respiration, and body temperature, (**C**) a combined, weighted feature set, (**D**) visual data, (**E**) performance data (accuracy and response time), (**F**) a combined, but unweighted feature set, and (**G**) a dummy classifier that considers only the distribution of the outcome classes for its prediction and represents an empirical chance level. Bootstrapped Bonferroni-corrected 95% confidence intervals (CI; 5000 iterations) of the mean F_1_ scores for the training set (left, orange) and the test set (right, blue) of the different unimodal and multimodal models. Notches in the boxes of the plot visualise the upper and lower boundary of the CI with the solid line representing the mean and the dashed grey line representing the median. The box comprises 50% of the distribution from the 25th to the 75th quartile. The ends of the whiskers represent the 5th and 95th quartile of the distribution. The continuous grey dashed line shows the upper boundary of the CI of the dummy classifier at 0.444.

Interestingly, for 8 out of 18 participants, we observed high prediction performances with F_1_ scores ranging between 0.7 and 1. However, we also identified several subjects whose subjectively perceived mental effort was hard to predict based on the training data of the other subjects. See Appendix A for a detailed comparison of the classifiers’ performances in the different test subjects. Concluding, the results indicate that transfer learning and generalisation over subjects is much more challenging when using the neurophysiological compared with the performance-based features.

### 3.2. Unimodal Predictions—Brain Activity

The unimodal voting classifiers for brain activity mainly used hard voting (94.4%) and gave the highest weights to the LDA classifier (Figure 6A). However, on average the unimodal voting classifier (M=0.40, 95% CI [0.31; 0.49], Figure 5A) revealed strong overfitting and was neither performing better than the single classifiers nor better than the estimated chance level. We then compared the performance of the classifiers with respect to the percentage of correctly and falsely classified cases in a confusion matrix (Figure 7). Therefore, we used the best-performing classifier for each test subject and then summed over all test subjects. We compared the distribution of the true positives, true negatives, false positives, and false negatives in these classifiers with the respective distribution of the voting classifier. Here (Figure 7A), we see that both distributions indicate a high number of falsely identified “High Mental Effort” cases (False Positives), leading to a recall of 45.6% and precision of only 39.3% for the voting classifier and a recall of 57.5% and precision of 49.8% for best single classifiers.

### 3.3. Unimodal Predictions—Physiological Measures

For classifying subjectively perceived mental effort based on physiological measures such as heart rate, respiration, and body temperature, soft voting was chosen in half of the test subjects. The weighting of the classifiers varied considerably, with the KNN obtaining the highest average weights (Figure 6B). The unimodal voting classifier (M=0.31, 95% CI [0.21; 0.40]; Figure 5B) showed strong overfitting, and its performance in the test subjects was neither significantly better than any of the single classifiers nor better than the estimated chance level. Regarding the percentage of correctly and falsely classified cases (Figure 7B), we see that the distributions for the best-performing single classifiers seem to be slightly better than the distributions of the voting classifier. The latter had difficulties in correctly identifying the conditions with low mental effort as can be seen in the high number of false negatives. When comparing the recall and precision of both approaches, we have a recall of only 29.9% for the voting classifier (precision: 38.6%) and an average recall of 51.0% for the best single classifiers (precision: 50.2%).

### 3.4. Unimodal Predictions—Ocular Measures

For subjectively perceived mental effort classification based on ocular measures such as pupil dilation and fixations, the split of soft vs. hard voting was 5.6% for soft voting and 94.4% for hard voting. KNN and SVM were weighted highest (Figure 6C). The F_1_ score of the unimodal voting classifier (M=0.35, 95% CI [0.26; 0.44]; Figure 5D) did not show a significant above-chance-level classification performance. The percentage of correctly and falsely classified cases (Figure 7D) was similar to the brain models, with a recall of 39.1% for the voting classifier (precision: 35.1%) and an average recall of 57.9% for the best single classifiers (average precision: 47.6%).

### 3.5. Unimodal Predictions—Performance

At last, we predicted subjectively perceived mental effort based on performance (accuracy and speed). 27.8% of the test subjects had voting classifiers using soft voting, and 72.2% used hard voting with SVM being weighted highest (Figure 6D). The models GNB (M=0.66, 95% CI [0.51; 0.79]), RFC (M=0.62, 95% CI [0.47; 0.75]), and SVM (M=0.64, 95% CI [0.47; 0.80]) showed all a significant above-chance-level performance (*p* < 0.01). The other models (LR, LDA, KNN) revealed also above-chance level performances but with smaller differences (*p*≈ 0.05; Table 2). The performance of the unimodal voting classifier (M=0.66, 95% CI [0.51; 0.79]) was also significantly better than the estimated chance level. The percentage of correctly and falsely classified cases (Figure 7E) reveals superior classification performance compared with the brain-, physiological- and ocular-based models. However, the voting classifier still had many falsely identified “High Mental Effort” cases (False Positives), leading to a recall of 78.5% and a precision of 57.6%. The best-performing single classifiers have an average recall of 82.4% and an average precision of 62.3%.

### 3.6. Unimodal Predictions Based on the Upper Quartile Split

To identify informative measures for very high perceived mental effort potentially reflecting cognitive overload, we also performed predictions based on the subject-wise split at the upper quartile. Compared with the median-split-based results, we observed decreased classifiers’ performance even below dummy classifier performance (Appendix A). This might be explained by the fact that we reframed a binary prediction problem with evenly distributed classes into an outlier detection problem. Using the upper quartile split, we created imbalanced classes regarding the number of the respective samples, which made the reliable identification of the less well-represented class in the training set more difficult (reflected in the recall; Appendix A).

### 3.7. Unimodal Predictions Based on the Experimental Condition

We further fitted models to predict the experimentally induced task load instead of the subjectively perceived mental effort. The prediction of mental effort operationalised by the task load was substantially more successful than the prediction of subjectively perceived mental effort. All modalities, including brain activity and physiological activity, revealed at least one classifier that was able to predict the current task load above the chance level (Appendix A). The unimodal voting classifiers (Brain: M=0.59, 95% CI [0.55; 0.62], Physiology: M=0.66, 95% CI [0.62; 0.72], Visual: M=0.69, 95% CI [0.62; 0.77], Performance: M=0.97, 95% CI [0.90; 1.0]) were all significantly better than a dummy classifier (M=0.51, 95% CI [0.47; 0.55]). Best unimodal voting classifications were obtained based on performance measures. Interestingly, other classification models were favoured in the unimodal voting, and the distribution between soft- and hard voting differed compared with the subjectively based approach, with soft voting being used more often (Appendix A).

### 3.8. Multimodal Predictions Based on the Median Split

In the final step, we combined the different modalities into a multimodal prediction. Figure 5C and Figure 7C show the performance of the multimodal voting classifier, and Figure 8A the average allocated weights to the different modalities. To compare the rather complex feature set construction of the multimodal voting with a simpler approach, we also trained two exemplary classifiers (LR without feature selection and RFC with additional feature selection) on the whole feature set without a previous splitting into the different modalities (Figure 5F).

In most test subjects (55.6%), soft voting was selected to combine the predictions for the different modalities; 44.4% used hard voting. In line with the results outlined above, the multimodal classifier relied on performance measures to predict subjectively perceived mental effort (Figure 8A), thereby turning it into a unimodal classifier. The voting classifier (Figure 5C, M=0.66, 95% CI [0.52; 0.80]) led to a significantly better classification than the estimated chance level. The multimodal classifier exhibited an equivalent percentage of correctly and falsely classified cases (Figure 7C) compared with the performance-based classifier, demonstrating an average recall of 78.5% and an average precision of 57.6%. On average, it performed better than the classifiers trained with the combined whole feature set, which showed substantial overfitting.

In order to assess the performance of the multimodal classifier without incorporating performance-based information such as speed and accuracy, we constrained the classifier to utilise only (neuro-)physiological and visual measures. This approach is especially relevant for naturalistic applications where obtaining an accurate assessment of behavioural performance is challenging or impossible within the critical time window. For the multimodal prediction without performance, brain activity was weighted highest (Figure 8B). However, classifiers revealed strong overfitting during the training, and the average performance was decreased to chance level (M=0.37, 95% CI [0.28; 0.46], average recall: 40.6% and average precision: 38.3%; Figure 5C).

### 3.9. Multimodal Predictions Based on the Upper Quartile Split

With the upper quartile split, we observed a fundamentally different allocation of weights. High weights were assigned to brain and ocular activity (Figure 8B), while performance received only minimal weights. Hence, the exclusion of performance-based measures had minimal impact on the allocation of weights (Appendix A) and the overall performance of the multimodal classifiers remained largely unaffected (Appendix A). Among the eighteen test subjects, the multimodal classification demonstrated the highest performance in two cases (Appendix A). However, on average, the multimodal classification based on an upper quartile split (M=0.19, 95% CI [0.08; 0.33]; average recall: 21.9% and average precision: 18.5%) did not demonstrate superiority over the unimodal classifiers (Brain: M=0.18, 95% CI [0.08; 0.30], Physiology: M=0.24, 95% CI [0.14; 0.34], Visual: M=0.16, 95% CI [0.06; 0.28], Performance: M=0.20, 95% CI [0.09; 0.34]). It further did not significantly outperform the dummy classifier (M=0.20, 95% CI [0.12; 0.28]) or classifiers trained on a feature set of simply combined modalities without weight assignment (LR: M=0.26, 95% CI [0.16; 0.36], RFC: M=0.28, 95% CI [0.18; 0.39]; Appendix A).

### 3.10. Multimodal Predictions Based on the Experimental Condition

Similar to the multimodal voting classifier based on a subject-wise median split of perceived mental effort, classifiers predicted the experimentally induced task load solely using the performance measures. The average prediction performance was exceptionally high (M=0.97, 95% CI [0.91; 1.0]; average recall: 99.7% and average precision: 91.3%), significantly outperforming a dummy classifier (M=0.51, 95% CI [0.47; 0.55]), and comparable to the performance of the classifiers trained on the combined feature set (LR: M=0.95, 95% CI [0.90; 0.99], RFC: M=0.96, 95% CI [0.91; 1.0]; Appendix A). When we only allowed (neuro-)physiological and visual measures as features, visual measures were weighted highest (Appendix A). In this case, the average performance of the multimodal classifiers (M=0.69, 95% CI [0.64; 0.74]) was also significantly above the chance level, with an average recall of 82.7% and precision of 58.7%, indicating a successful identification of mental effort based on neurophysiological, physiological, and visual measures (Appendix A).

## 4. Discussion

The purpose of our study was to test the feasibility of multimodal voting in a ML classification for complex close-to-realistic scenarios. We used both—the subjectively experienced and experimentally induced mental effort—as ground truths for a cross-subject classification. Our approach represents a crucial investigation for the practical application of mental state decoding under real-world conditions. Our study aims to fill the existing gap in the literature and address the need for online accessible naturalistic data sets by providing a multimodal voting ML architecture along with the dataset to decode mental effort across subjects in a quasi-realistic experiment simulating a real-world monitoring task. This serves as the foundation for enabling the adaptation of systems to users’ current mental resources and efforts. By incorporating adaptive systems, individuals can enhance their performance by operating within an optimal level of demand, allowing them to perform at their best. In tasks involving high-security risks, it is crucial for system engineers to make every effort to prevent individuals from being overwhelmed or bored, as such states can increase the likelihood of errors. In our analyses, we employed multimodal voting cross-subject classification and evaluated the model performance using a leave-one-out approach. We contribute to the existing body of knowledge on mental state decoding by systematically evaluating the selection and informativeness of sensors including neurophysiological, physiological, visual, and behavioural measures for the classification of subjective and experimentally induced mental effort. Moreover, our results show which classifier models perform best for each modality. We further observed that in certain modalities, the combination of ML models outperformed predictions made by individual ones. For each modality, we found a different set of classifiers that were better performing in the prediction and, thus, also considered more informative in the unimodal voting.

### 4.1. Using Subjectively Perceived Mental Effort as Ground Truth

When predicting subjectively perceived mental effort, LDA and LR performed best and were weighted highest in the classifications based on brain activity (Figure 5). Whereas, in physiological activity, the highest weights were assigned to KNN, RFC, and SVM. Regarding visual activity, the GNB and KNN revealed high classification performance among the test subjects. However, these models aiming to predict subjectively perceived mental effort based on brain activity, physiological activity, and visual measures were still strongly overfitted, and their performances in the test subjects were not significantly better than the dummy classifier. In performance-related measures, the GNB, RFC, and SVM performed significantly better than the dummy classifier when predicting subjective mental effort based on a median split. Using the upper quartile split for performance-related measures, the KNN and SVM showed the highest, but still, chance-level-like performances. Regarding the unimodal voting predictions of subjectively perceived mental effort, we see that a weighted combination of classifiers (LR, LDA, GNB, KNN, RFC, and SVM) was not superior to single classifiers neither when using the median nor the upper quartile split. When we combined the different modalities into a joined prediction of subjectively perceived mental effort, only the performance modality was considered. Hence, our multimodal classification might rather be considered a unimodal (performance-based) prediction. Removing the performance information from the multimodal voting classifier increases overfitting and drops the average classification performance. However, a more detailed investigation of the upper quartile split classification revealed that performance was less predictive in identifying cases of exceptionally high perceived mental effort and potential “cognitive overload” (Figure 7). In the upper quartile split classification, higher multimodal voting weights were assigned to neurophysiological and visual measures compared with performance measures. This seems to imply that subjects were more heterogeneous in their performance under exceptionally high perceived mental effort, and classifiers rather exploited correlates from neurophysiological and visual measures than from performance to predict subjectively perceived mental effort. In summary, our findings indicate that when utilizing only unimodal voting classification, the best prediction of subjectively perceived mental effort was achieved through performance-based measures. Additionally, the inclusion of the performance-based classification model is essential in our multimodal voting classification approach to address potential overfitting in predicting mental effort (median-based split). These findings suggest that further research is necessary to investigate the dependence and variability of mental effort in cross-subject classification.

### 4.2. Using Experimentally Induced Mental Effort as Ground Truth

For the classification of the experimentally induced task load, all modalities were able to predict mental effort with high performances already on a single classifier level. GNB, KNN, and SVM performed above the chance level and were assigned the highest weights in the unimodal voting based on brain activity (Appendix A). For the physiological activity, all classifiers—except the KNN—reached above-chance level performance. The highest average weight in the unimodal voting was assigned to the SVM. For visual and performance measures, we did not see substantial differences between the classification performance of the single models, with all performing above the chance level. A unimodal weighted combination of these classifiers was not superior to the single classifiers in any modality. Performance exhibited the highest predictive capability for task load (Appendix A). As a result, the multimodal classifier transitioned again back into unimodal voting, as it relied solely on the performance modality. When excluding the performance-based features, the multimodal prediction based on neurophysiological, physiological, and ocular activity was still significantly above the chance level estimated by a dummy classifier. These findings suggest that it is feasible to differentiate between various mental effort states, represented by experimentally induced task load, by utilizing neurophysiological, physiological, and visual data obtained in a close-to-realistic environment through a cross-subject classification approach. However, it was not possible to replicate these results for subjectively perceived mental effort. The discrepancies observed between these two ground truth approaches could potentially be attributed to the retrospective nature of self-reports. Self-reports rely on an individual’s perception, reasoning, and subjective introspection [79]. They are, therefore, vulnerable to various perceptual and response biases like social desirability [21,80]. These post hoc evaluation processes might not be adequately reflected in and could be learned from (neuro-)physiological and visual measures during the task itself.

### 4.3. Generalisation across Subjects

For all classification approaches, we observed substantial variation in the performance of classifiers between the test subjects. Some individuals had F_1_ scores above 0.8 (Appendix A). Other individuals demonstrated deviations in their neurophysiological reactions, diverging significantly from the patterns learned from the subjects included in the training set. These results are in line with the findings by Causse et al. [81]. The authors concluded that it is quite challenging to identify mental states based on haemodynamic activity across individuals because of the major structural and functional inter-individual differences. For instance, in the context of brain–computer interfaces, a phenomenon called BCI illiteracy describes the inability to modulate sensorimotor rhythms in order to control a BCI observed in approximately 20–30% of subjects [82]. Our findings underscore the importance of developing appropriate methods to address two key aspects. First, identifying subjects who may pose challenges in prediction due to their heterogeneity compared with the training set. Second, enabling transfer learning for these individuals by implementing techniques such as standardisation and transformation of correlates into a unified feature space [83].

### 4.4. Limitations and Future Research

We acknowledge that certain aspects of this study can be further improved and serve as opportunities for future advancements. One area for improvement is the complexity of the measurement setup used in this study, which required a substantial amount of time for the preparation and calibration of the involved devices. It is important to consider the potential impact on participants’ intrinsic engagement and explore ways to further streamline the process during soft- and hardware development. Furthermore, it is worth noting that our feasibility study sample was relatively small and homogeneous in terms of socio-demographic characteristics, consisting predominantly of young individuals with a high level of education. The small sample size likely had a negative impact on the statistical power of our study. Combined with the homogeneity of the sample, it could also limit the generalisability of our results to more diverse populations. While it may seem intuitive to increase the sample size to address the issue of heterogeneity, there is a debate surrounding the relationship between sample size and its impact on classifier performance. With adding more and more samples, the dataset is supposedly at some point large enough to enable the classifier to find more generic and universal predictive patterns and achieve better performance again. Some argue that it is necessary to train ML models with large training datasets, including edge cases, to achieve good generalisability and attain good prediction accuracy on an individual-level Bzdok and Meyer-Lindenberg [84], Dwyer et al. [85]. Nevertheless, as emphasised by Cearns et al. [86], it is worth noting that ML classifiers demonstrate exceptional performance primarily in relatively small datasets. Consequently, the heterogeneity of a large dataset might present a significant challenge for learning. Thus, it may be more reasonable to train separate, specialised models for each homogeneous cluster, rather than attempting to construct a single model that explains the entire variance but yields less accurate predictions. Orrù et al. [87], for example, suggests the use of simple classifiers or ensemble learning methods instead of complex neural networks. Cearns et al. [86] highlight the importance of suitable cross-validation methods. Especially in the case of physiological datasets, one might also identify subjects that are very predictive for the patterns of a specific subgroup and remove subjects from the training set that show unusual patterns in neurophysiological reactions [85]. One interesting idea to address this problem is data augmentation [88,89]. This can be done by artificially generating new samples from existing samples to extend a dataset. For example, using Generative Adversarial Networks (GANs), one could simulate data to create more homogeneous and “prototypic” training datasets and increase the performance and stability of respective ML models [90]. Another suggested method to improve generalisability across subjects might be multiway canonical correlation analysis (MCCA). An approach that allows combining multiple data sets into a common representation and, thereby, achieves the denoising of data, and dimensionality reduction, based on shared components across subjects [83]. Advancements in these methods play a crucial role in enhancing the comparability and potential combinability of datasets, which is a shared objective within the research community [91].

To further increase classification performance, additional artefact analyses [92], or the implementation of inclusion criteria on a subject-, trial-, and channel-level could be explored in order to improve poor signal-to-noise ratios. Friedman et al. [93], who used an XGBoost classifier on EEG data, applied extensive and rigorous trial and subject selection criteria. For example, they did not include trials where participants failed to solve the task because they assumed that the mental effort shown by participants answering incorrectly did not reflect the true level of load (also [94]). Although this bears the risk of a major data loss, these rigid removal criteria might reflect an efficient solution to ensure that the measured neurophysiological signals truly reflect the cognitive processes of interest. Future research is necessary to a) define such exclusion and inclusion criteria depending on the investigated cognitive processes and b) develop standardised evaluation methods to decide which preprocessing step is beneficial and adequate.

A final limitation relates to the arrangement of the fNIRS optodes. Based on previous research (e.g., [12]), we decided to choose a montage solely covering the prefrontal cortex in order to reduce preparation time and facilitate transfer into close-to-realistic applications. However, we probably would have profited from a larger brain coverage that also covers parietal, temporal, and occipital brain areas [94]. Integrating these regions allows identifying features for the classification from larger functional networks that might play a crucial role in distinguishing mental states and cognitive control mechanisms [36,95]. Increased activity in the frontoparietal network is, for example, associated with task-related working memory (WM) processes (e.g., [96,97]), whereas increased connectivity between frontal and sensory areas are linked to the suppression of distractors [95].

### 4.5. Feature Selection and Data Fusion in Machine Learning

A crucial aim of this study was the selection and fusion of informative sources for cross-subject mental effort prediction. We integrated data from different modalities comprising brain activity as assessed with fNIRS, physiological activity (cardiac activity, respiration, and body temperature), ocular measures (pupil dilation and fixations), as well as behavioural measures of performance (accuracy and speed). However, this selection was naturally not exhaustive. Other measures, such as electroencephalography or electrodermal activity [98], could provide useful information about cognitive and physiological processes related to mental effort. In addition, one could also explore more behaviour-related measures such as speech [99] or gaze [100]. These measures might also provide the possibility to detect predictive patterns without significantly interfering with the actual task. When conducting applied research and incorporating mental state decoding in real-world settings (e.g., healthcare, entertainment, gaming, industry, and lifestyle; [101]), it is crucial to utilise sensors that are unobtrusive, seamlessly integrated into the environment, mobile, and user-friendly. For this purpose, further validation studies are warranted to evaluate the quality and suitability of smart wearables such as smartwatches or fitness trackers [101,102], mobile neurophysiological sensors [103,104], and mobile eye-tracking [105,106]. Our results indicate that performance-based measures as well as a multimodal approach including neurophysiological, physiological, and visual measures are successful in decoding experimentally induced mental effort. Visual features, followed by physiological measures, were particularly informative in the multimodal approach. This insight enables researchers to optimise their sensor setup by prioritizing measures and streamlining their data collection process.

To combine the data streams obtained from the different measurement methods, we implemented data fusion on two levels: (1) the feature level and (2) the classification level. First, we aggregated our raw data, mainly time series, into informative features. We used standard statistical features like the mean, standard deviation, skewness, and kurtosis. Friedman et al. [93] explored more sophisticated features such as connectivity and complexity metrics, which have the potential to capture additional information about relationships within and between neuronal networks. Further investigations are required to assess the predictive quality of these aggregated features. Additionally, future research can explore the added value of feature selection and wrapping methods, which aim to reduce the complexity of the feature space without compromising the predictive information [107,108]. Such methods, e.g., sequential feature forward selection, might be a way to improve classifiers’ performance by keeping only the most informative features. Another approach could be the use of continuous time-series data which provide insights into differences in the experience and processing of mentally demanding tasks separately for the different neurophysiological modalities. Hence, some researchers implemented deep learning methods like convolutional or recurrent neural networks to derive classifications based on multidimensional time-series data [45,109,110]. Nevertheless, these algorithms require that all data streams are complete (no missing data points) and have the same length and sampling frequency. These requirements are often difficult to fulfil in naturalistic settings with multimodal measurement methods using different measurement devices.

Once the feature space is defined, the research focus shifts towards developing strategies for selecting, merging, combining, and weighting multiple classifier models and modalities at the classification level. These strategies are still the subject of ongoing research and exploration. In this context, it is important to strike a balance between computational power, dataset size, and the benefits of finely tuned combinations of optimally stacked or voted classifiers. The exploration of early and late fusion approaches, as commonly employed in the field of robotics, could provide valuable insights. Early fusion involves the early combination of all data points and the fitting of classifiers to multidimensional data. On the other hand, late fusion involves a more fine-grained pipeline, where several classifiers are fitted to different proportions of the dataset and subsequently combined at a later stage. In this study, we implemented a late-fusion approach where we first combined different classifiers for each modality. In a subsequent step, we combined classifiers to create a unified prediction. Exploring early and late fusion strategies is especially important when one wants to account for temporal dynamics in the different measures or the realisation of real-time mental state monitoring. The review of Debie et al. [27] provides a comprehensive overview of the different fusion stages when identifying mental effort based on neurophysiological measures.

## 5. Practical Implications and Conclusions

Our proposed multimodal voting classification approach contributes to the ecologically valid distinction and identification of different states of mental effort. It paves the way toward generalised state monitoring across individuals in realistic applications. Interestingly, the choice of ground truth had a fundamental influence on the classification performance. The prediction of subjectively perceived mental effort operationalised through self-reports, is most effectively achieved by incorporating performance-based measures. On the other hand, the experimentally induced task load can be accurately predicted not only from performance-based measures but also by incorporating neurophysiological, physiological and visual measures. Our findings provide valuable guidance for researchers and practitioners in selecting appropriate methods based on their specific research questions or application scenarios, taking into account limited resources or environmental constraints. The capacity to predict subjectively perceived and experimentally induced mental effort on an individual level makes this architecture an integral part of future research and development of user-centred applications such as adaptive assistance systems.

## Figures and Tables

**Figure 1 sensors-23-06546-f001:**
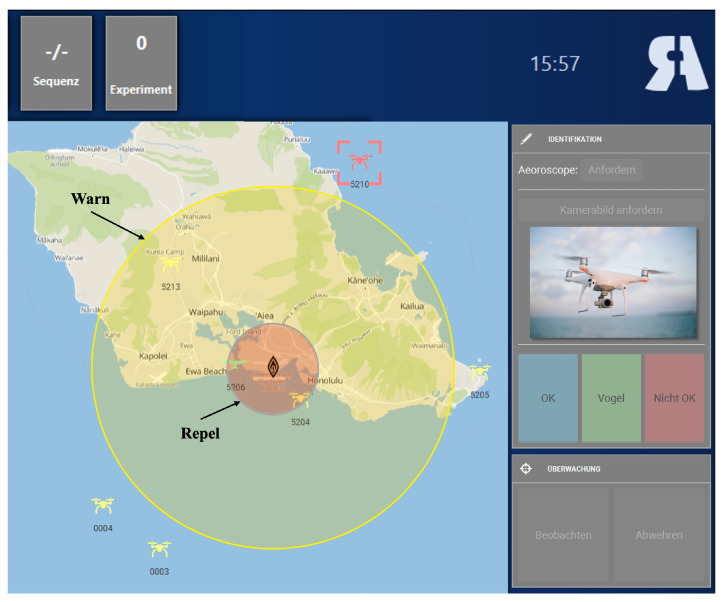
Elements of the WCT interface. Left side of the screen (map): Participants had to monitor the aerial space of the airport. When an unregistered drone entered the yellow area (outer circle), participants had to warn that drone; when an unregistered drone entered the red area (inner circle), participants had to repel it. Right side of the screen (graphical user interface): Participants had to request codes and pictures of unknown flying objects and then classify them as birds, registered drones, or unregistered drones.

**Figure 2 sensors-23-06546-f002:**
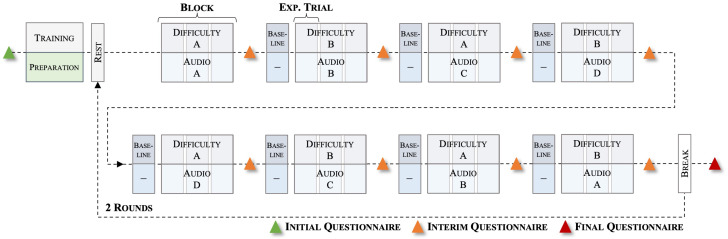
Procedure of the experiment. The presented procedure is exemplary as the task load condition was alternating, and the concurrent emotional condition was pseudo-randomised throughout the different blocks.

**Figure 3 sensors-23-06546-f003:**
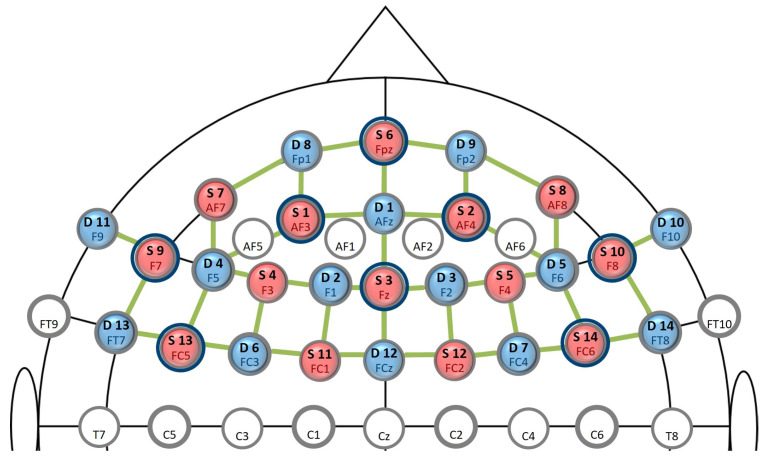
Location of fNIRS optodes. Montage of optodes on fNIRS cap on a standard 10–20 EEG system, red optodes: sources, blue optodes: detectors, green lines: long channels, dark blue circles: short channels. Setup with 41 (source–detector pairs) × 2 (wavelengths) = 82 optical channels of interest.

**Figure 4 sensors-23-06546-f004:**
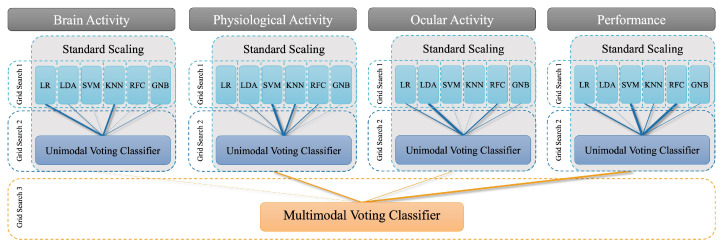
Classification procedure with cross-validated randomised grid searches (maximum number of 100 iterations) and a validation set consisting of one or two subjects. The first grid search optimises the hyperparameters for the different individual and unimodal classifiers. The second grid search optimises the weights as well as voting procedure (soft or hard) for the unimodal voting classifier. The third grid search optimises the weights as well as the voting procedure (soft or hard) for the multimodal voting classifier.

**Figure 6 sensors-23-06546-f006:**
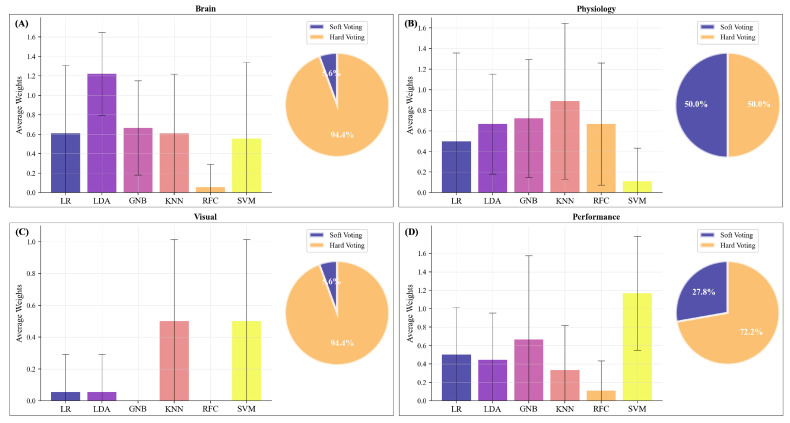
Weights and procedure of an unimodal voting classifier to predict subjectively perceived mental effort based on a median split; validation set: N=1. Allocated weights for an unimodal voting classifier based on (**A**) fNIRS data, (**B**) physiological data (heart rate, respiration, and body temperature), (**C**) visual data, and (**D**) performance data (accuracy and response time). Error bars represent the standard deviation.

**Figure 7 sensors-23-06546-f007:**
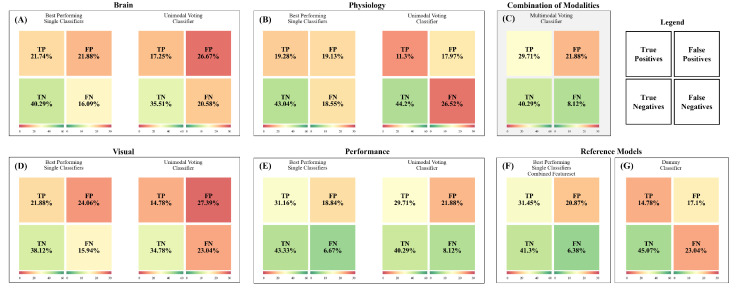
Prediction of the subjectively perceived mental effort (confusion matrix of test set) based on a median split; validation set: N=1. Percentage of correctly and falsely classified perceived mental effort per model across all test subjects: TP = True Positives, TN = True Negatives, FP = False Positives, and FN = False Negatives, with “Positives” representing “High Mental Effort” and “Negatives” representing “Low Mental Effort”. For the “Best Performing Single Classifier” we selected the classifier (LDA, LR, SVM, KNN, RFC, or GNB) with the best F_1_ score for each subject. Confusion matrices based on (**A**) fNIRS data, (**B**) physiological data (heart rate, respiration, and body temperature), (**C**) a combined, weighted feature set, (**D**) visual data, (**E**) performance data (accuracy and response time), (**F**) a combined, but unweighted feature set, and (**G**) a dummy classifier representing an empirical chance level.

**Figure 8 sensors-23-06546-f008:**
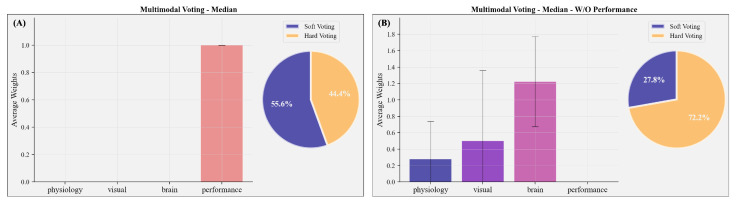
Weights and procedure of a multimodal voting classifier to predict subjectively perceived mental effort based on a median split; validation set: N=1. (**A**) shows the allocation of weights when all modalities are included in the multimodal classification. (**B**) shows the allocation of weights when performance measures are not included in the multimodal classification. Error bars represent the standard deviation.

**Table 1 sensors-23-06546-t001:** Included features per modality.

Modality	Features
**Brain Activity**	Mean, standard deviation, peak-to-peak (PTP) amplitude, skewness, and kurtosis of the 82 optical channels
**Physiology**	
Heart Rate	Mean, standard deviation, skewness, and kurtosis of heart rate
	Mean, standard deviation, skewness, and kurtosis of heart rate variability
Respiration	Mean, standard deviation, skewness, and kurtosis of respiration rate
	Mean, standard deviation, skewness, and kurtosis of respiration amplitude
Temperature	Mean, standard deviation, skewness, and kurtosis of body temperature
**Ocular Measures**	
Fixations	Number of fixations, total duration and average duration of fixations, and standard deviation of the duration of fixations
Pupillometry	Mean, standard deviation, skewness, and kurtosis of pupil dilation
**Performance**	Average reaction time and cumulative accuracy

**Table 2 sensors-23-06546-t002:** Bootstrapped Bonferroni-corrected means and 95% CIs of F_1_ scores.

	Training Set	Test Set
Chance Level		
Dummy Classifier	0.351, 95% CI [0.320; 0.379]	0.368, 95% CI [0.284; **0.444**]
Unimodal Predictions Based on fNIRS		
LR	0.924, 95% CI [0.889; 0.962]	0.392, 95% CI [0.278; 0.500]
LDA	0.845, 95% CI [0.840; 0.850]	0.387, 95% CI [0.271; 0.495]
GNB	0.532, 95% CI [0.469; 0.577]	0.366, 95% CI [0.275; 0.457]
KNN	1.0, 95% CI [1.0; 1.0]	0.348, 95% CI [0.248; 0.451]
RFC	0.721, 95% CI [0.667; 0.786]	0.308, 95% CI [0.235; 0.387]
SVM	0.756, 95% CI [0.644; 0.862]	0.386, 95% CI [0.287; 0.473]
Unimodal Voting	0.911, 95% CI [0.848; 0.954]	0.401, 95% CI [0.314; 0.489]
Unimodal Predictions Based on Physiology		
LR	0.441, 95% CI [0.426; 0.454]	0.354, 95% CI [0.246; 0.464]
LDA	0.441, 95% CI [0.424; 0.456]	0.341, 95% CI [0.226; 0.448]
GNB	0.279, 95% CI [0.269; 0.297]	0.231, 95% CI [0.141; 0.318]
KNN	0.966, 95% CI [0.962; 0.973]	0.377, 95% CI [0.309; 0.439]
RFC	0.648, 95% CI [0.585; 0.708]	0.327, 95% CI [0.217; 0.425]
SVM	0.532, 95% CI [0.379; 0.715]	0.366, 95% CI [0.262; 0.455]
Unimodal Voting	0.767, 95% CI [0.691; 0.848]	0.305, 95% CI [0.212; 0.396]
Unimodal Predictions Based on Visual Measures		
LR	0.318, 95% CI [0.287; 0.355]	0.201, 95% CI [0.080; 0.354]
LDA	0.314, 95% CI [0.285; 0.345]	0.198, 95% CI [0.087; 0.346]
GNB	0.492, 95% CI [0.472; 0.506]	0.390, 95% CI [0.294; 0.485]
KNN	1.0, 95% CI [1.0; 1.0]	0.386, 95% CI [0.291; 0.485]
RFC	0.690, 95% CI [0.658; 0.728]	0.322, 95% CI [0.237; 0.415]
SVM	0.517, 95% CI [0.422; 0.630]	0.301, 95% CI [0.198; 0.401]
Unimodal Voting	0.751, 95% CI [0.589; 0.915]	0.354, 95% CI [0.262; 0.442]
Unimodal Predictions Based on Performance Measures		
LR	0.586, 95% CI [0.549; 0.624]	**0.543**, 95% CI [0.399; 0.676] *
LDA	0.584, 95% CI [0.548; 0.618]	**0.499**, 95% CI [0.325; 0.663] *
GNB	0.667, 95% CI [0.659; 0.677]	0.661, 95% CI [**0.514**; 0.792] **
KNN	0.919, 95% CI [0.914; 0.924]	**0.513**, 95% CI [0.367; 0.650] *
RFC	0.696, 95% CI [0.686; 0.707]	0.616, 95% CI [**0.472**; 0.752] **
SVM	0.679, 95% CI [0.672; 0.689]	0.641, 95% CI [**0.467**; 0.799] **
Unimodal Voting	0.673, 95% CI [0.662; 0.687]	0.656, 95% CI [**0.509**; 0.789] **
Multimodal Predictions		
LR	0.919, 95% CI [0.892; 0.939]	**0.498**, 95% CI [0.412; 0.581] *
RFC	0.803, 95% CI [0.784; 0.830]	0.617, 95% CI [**0.499**; 0.742] **
Multimodal Voting	0.673, 95% CI [0.663; 0.686]	0.658, 95% CI [**0.515**; 0.797] **
Multimodal Voting (w/o Perf.)	0.930, 95% CI [0.905; 0.957]	0.369, 95% CI [0.276; 0.464]

Note. * *p* < 0.05 considering the bootstrapped mean, ** *p* < 0.01 considering the lower CI limit.

## Data Availability

The datasets analysed for this study as well as the code can be found in a publicly accessible OSF repository: https://osf.io/9dbcj/, accessed on 16 July 2023.

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
