# Peer review of "Decoding Mental Effort in a Quasi-Realistic Scenario: A Feasibility Study on Multimodal Data Fusion and Classification"

_sensors, 2023, doi:10.3390/s23146546_

Round 1

Reviewer 1 Report

The study investigates the feasibility of decoding mental effort from multimodal physiological and behavioural signals in a quasi-realistic scenario. The topic is of great interest and the paper is well written. In my opinion, only few concerns need to be addressed before publication.

1)     In the manuscript some acronymous are disclosed not only at the first appearance (e.g., PFC is explained in line 37 and 82, HbO in line 36 and 250)). Please check the manuscript and explain the manuscript only at the firs appearance and use the acronym along the manuscript.

2)     In the manuscript it is stated that some models perform (or not) significantly better than other models, however, in my opinion, a statistical comparison among the accuracies of the different developed models should be reported. Please, add a table to report the p-values of the comparisons between the different models.

3)     The multimodal approach indeed outperforms the unimodal one when predicting cognitive workload. However, the Authors should discuss about the feasibility to employ the variety of sensors used in this study in ecological applications. In this perspective, the Authors could think to develop in further studies multimodal approaches based on contactless or wearables (e.g., smartwatches) to monitor the mental effort of individuals. Please describe this aspect in the Discussion section.

Author Response

                                                                                       Stuttgart, July 14th 2023

Dear Reviewer 1,         

We thank you for evaluating our manuscript sensors-2479390 for publication as a Regular Research Article in the Journal Sensors - Special Issue "Sensors for Human Activity Recognition II".

»Decoding Mental Effort in a Quasi-Realistic Scenario: A Feasibility Study on Multimodal Data Fusion and Classification«

We are delighted to receive the opportunity to publish our revised manuscript.

We appreciate the time and effort that you dedicated to providing feedback on our manuscript and are grateful for the insightful comments and valuable improvements to our paper.

We have incorporated the suggestions for improvement into the manuscript. Those changes are highlighted in yellow in the manuscript. Please see the attachment for a point-by-point response to the your comments and concerns. Line numbers refer to the revised manuscript file.

We believe that the revision has contributed greatly to the clarity of the manuscript and would like to thank the reviewers for their efforts.

Sincerely,

Katharina Lingelbach

Reviewer 2 Report

Dear Editor

Gado et al,  tested the feasibility of multimodal voting in a machine learning classification for complex close-to-realistic scenarios.

The study is interesting however it needs some clarifications:

The inclusion criteria must be clear.

Why was the sample size  small? The limitation of the study must be mentioned.

The exclusion criteria must be refined and organized, the usage of and/or is not clear.

Was each participant tested individually or they all were situated in the same place?

In the purpose section, the novelty of the study is not clear. What this research added to the previous similar ones?

A flowchart of the applied method and a summary of results could make the comprehension much easier, So I suggest to add a graphical abstract of the study.

Author Response

                                                                                           Stuttgart, July 14th 2023

Dear Reviewer 2,       

We thank you for evaluating our manuscript sensors-2479390 for publication as a Regular Research Article in the Journal Sensors - Special Issue "Sensors for Human Activity Recognition II".

»Decoding Mental Effort in a Quasi-Realistic Scenario: A Feasibility Study on Multimodal Data Fusion and Classification«

We are delighted to receive the opportunity to publish our revised manuscript.

We appreciate the time and effort that you dedicated to providing feedback on our manuscript and are grateful for the insightful comments and valuable improvements to our paper.

We have incorporated the suggestions for improvement into the manuscript. Those changes are highlighted in yellow in the manuscript. Please see see the attachment for a point-by-point response to the comments and concerns. Line numbers refer to the revised manuscript file.

We believe that the revision has contributed greatly to the clarity of the manuscript and would like to thank you for your efforts.

Sincerely,
Katharina Lingelbach
